# Antiproliferative and Antimicrobial Activity of Anthocyanins from Berry Fruits after Their Isolation and Freeze-Drying

**Ivan Salamon** [1,*] , **Ela Nur Şimşek Sezer** [2], **Maryna Kryvtsova** [3] **and Pavol Labun** [4]

1 Department of Ecology, Faculty of Humanities and Natural Science, University of Presov,
  081 16 Presov, Slovakia
2 Department of Biology, Faculty of Science, Selçuk University, Konya 42130, Turkey; elasimsek@selcuk.edu.tr
3 Department of Genetics, Plant Physiology and Microbiology, Uzhhorod National University
  Biological Faculty, 88000 Uzhhorod, Ukraine; maryna.krivcova@uzhnu.edu.ua
4 Pharma Bavaria, Hochriensstr. 36, 83209 Prien am Chiemsee, Germany; lianko@azet.sk
* Correspondence: ivan.salamon@unipo.sk

**Abstract:** Natural phytonutrients in foods, including anthocyanins, can play an important role in human health. Anthocyanins have been reported to cause many various useful effects, such as reducing cancer cell proliferation, regulating blood pressure, preventing tumor formation, and preventing diabetes. In this study, we aimed to reveal the qualitative anthocyanin content, antiproliferative and antimicrobial effects of different extracts derived from *Vaccinium myrtillus*, *Vaccinium corymbosum*, *Sambucus nigra* and *Aronia melanocarpa*. The anthocyanin content of the plants mentioned in the study was characterized after the freeze-drying process. MTT assay was chosen to determine the antiproliferative effect of extracts on colorectal cancer cells. Antimicrobial effects of extracts were studied on typical and clinical strains of five different bacteria. As a result, it was determined that the anthocyanin content in the extracts obtained by the freeze-drying method was acceptable, and some extracts were found to have a strong antiproliferative effect on colorectal cancer cells and have an antibacterial effect on typical and clinical strains. Conclusionally, it has been shown that anthocyanins, which are reported to have many beneficial effects, are very promising when prepared with a relatively new method, and this study is a pioneer study for possible future pharmacological and in vivo studies.

**Keywords:** anthocyanins; antimicrobial; berries; cytotoxic; extraction; lyophilization

## 1. Introduction

Food Science is a multidisciplinary field that provides scientific knowledge to solve problems associated with many aspects of the food system. The basis of this discipline is an understanding of the chemistry of food ingredients and the changes they undergo during processing and storage. It is important to fully understand the methods of processes such as drying, freezing and pasteurization.

Phytonutrients are found in plants and have certain biological properties. Some important bioactive phytonutrients, which also have specific pharmacological effects [1], include terpenoids, anthocyanins, flavonoids, isoflavonoids, and polyphenols.

Anthocyanins are one of the most common natural pigments in the plant kingdom. They are water-soluble pigments responsible for the colours of plant foods ranging from red to purple and have a positive effect on human health. They are very sensitive to changing of pH and temperature. Anthocyanins are flavonoid substances belonging to the group of polyphenols. There are about 7,000 different flavonoids found in most fruits, vegetables, and various drinks (e.g., juice, beer, wine, and coffee). Each plant species has a characteristic anthocyanin pigment that exhibits a range of colours from red to blue, hence bearing the colours of fruits and vegetables. Anthocyanins are found in nature in the form of O-glycosides, their aglycones (non-saccharide part) are called anthocyanidins.

The difference between the individual aglycon types depends on a number of linked hydroxylated groups and their methylation level. The most common anthocyanidins are cyanidine, delphinidine, peonidine, malvidine, pelargonidine and petunidine.

At the same time, fruits and phytonutrients have an important role in the prevention and treatment of various diseases. The beneficial effects of anthocyanins on human health have been known since the 16th century. For example, the use of blackberries to treat eye and mouth infections is very old [2]. Moreover, natural anthocyanins have been extensively studied for their medicinal potential. Anthocyanins have protective effects on cardiovascular diseases as well as anti-inflammatory, anticancer, antimicrobial, anti-obesity and antidiabetic effects [3]. The antioxidant properties of anthocyanin extracts from fruits of various plant species have been the subject of several research works [4–6]. The results indicated that all berry samples had a good antioxidant activity that depends on several factors, such as plants genotype, soil-climatic properties, various anthocyanin isolation and test (DPPH$^+$, ABTS$^+$, TEAC) methods. Based on total phenolic compounds concentration (from 3567 mg/100 g to 1050 mg/100 g) and antioxidant capacities (from 84.5% to 62.8% of radical scavenging capacity) the investigated berry species may be ranked as follows: *Aronia melanocarpa* Wild. > *Sambucus nigra* L. > *Vaccinium myrtillus* L. > *Vaccinium corymbosum* L. The IC$_{50}$ values and the Trolox equivalent antioxidant capacity (TEAC) values were the highest for *Aronia melanocarpa* ethanolic extract, while the acetone sample prove to be nine time less active.

Cancer is an insidious disease with a high mortality rate in the human population. Cancer cells differ from normal cells in terms of morphology and function. Anthocyanins act on cancer cells due to these differences. The most important feature of cancer cells is the loss of control over cell division and subsequent continuous proliferation. Anthocyanin-rich extracts and pure anthocyanins have been reported to inhibit cell proliferation by blocking various stages of the cell cycle in cancer cells [7,8]. Previous studies have reported anti-proliferative effects of the anthocyanin-rich extracts on different cancer cell lines such as HT-29, MCF-7, HEp-2, MKN-45, HL-60 and A549 [9,10]. Despite the data obtained, there is still little information about the synergistic or antagonistic effects of various anthocyanins on the progression and inhibition of the carcinogenesis process, which needs to be clarified by much more in vitro and in vivo studies.

The anthocyanin properties mentioned above are the main reason for the increased interest in studies in this field in recent years. Therefore, anthocyanins derived from edible plants can potentially be considered as pharmaceutical ingredients. Among different plant species, berry fruits are rich in various polyphenols, especially anthocyanins. The anthocyanin content of strawberry fruits ranges from 7.5 mg/100 mg of fresh fruit in currant (*Ribes rubum*) to 460 mg/100 g of fresh fruit in chokeberry (*Aronia melanocarpa*) [11]. Today, new ways of preserving various biological properties of anthocyanins are sought in the pharmacology and food industries. Freeze drying is considered a suitable and relatively new method for drying heat sensitive pigments. It is very modern with its freeze-drying process, water extraction and product stabilization capability. This process is based on dehydration of a frozen product by sublimation, and during this procedure, core materials and matrix solutions are homogenized and then lyophilized to obtain a dry material [12,13].

The purpose of our study is to lyophilize pure anthocyanins stabilized in powder form to vials with different solvents and to perform qualitative and quantitative characterization of anthocyanins isolated from selected plant species after freeze-drying. In addition, we aim to reveal the various biological activities of anthocyanins obtained from different berry fruits, such as antiproliferative and antimicrobial effects.

## 2. Materials and Methods

### 2.1. Fruit Source

High-bush blueberry, *Vaccinium corymbosum* L.,
Bilberry, *Vaccinium myrtillus* L.,
Elderberry, *Sambucus nigra* L.

Black Chokeberry, *Aronia melanocarpa* Wild.

Fresh fruits were obtained from local farms in Eastern Slovakia. Samples were stored at temperature of −20 °C until use.

### 2.2. Chemicals

Ethanol, acetone, hydrochloric, oxalic, citric or siccine acids as well as adsorbents (Amberlite XAD-7, Talcum and C18) were purchased from Sigma Aldrich and Merck (4, Dvorakovo nabr., Bratislava, Slovakia). Anthocyanin standards for comparison were supplied by Polyphenols Laboratories AS (4-6, Hanaveien, 4327, Sandnes, Rogalan, Norway). All used chemicals were analytical grade.

### 2.3. Preparation of Plant Extracts

**Ethanol extraction of anthocyanins:** Fresh fruits were blended in home blender. Samples with weight of 1000 g fruit were mixed with double (weight to volume) volume of 20–96% (volume to volume) ethanol-water solution acidified by 0–5% of hydrochloric, oxalic, citric or siccine acids for 0.5–1.5 h extraction with continuous mixing. Extracts were separated from material by filtration through a filter paper with vacuum suction using a Buchner funnel and water-flow pump. The material was mixed with fresh ethanol-water solution two more times for maximal extraction of anthocyanins. Filtrates were moved to boiling flask and ethanol was removed by rotary evaporator. Purification was carried out by mixing of 30–50 g of solid adsorbents (Amberlite XAD-7, Talcum and C18) which was activated with double volumes of ethanol and then with three volumes of acidified deionized water. Filtrate was mixed with 50 g of adsorbent and separated from filtrate by filtration through a filter paper. Adsorbent was flushed by two volumes of acidified water with 1% citric acid (to remove water soluble compounds—colorants, sugars, organic acids etc.) and that with two volumes of ethyl acetate (to remove polyphenols). Elute anthocyanin pigment was removed from solid adsorbent by extraction with acidified ethanol-water solution. Ethanol was removed by vacuum evaporation at 38 °C.

**Acetone extraction of anthocyanins:** 1000 g of material was macerated with the same amount (weight to volume) of acetone. Filtrate was separated by vacuum suction using filter paper and Buchner funnel. Extraction of anthocyanins was carried out by maceration of fruit with 70% ($v/v$) aqueous acetone (70% of acetone and 30% of 1% citric acid water solution) for 30 min. Filtrate was separated by filter paper and Buchner funnel, moved to a separatory funnel and mixed with double volume of chloroform by turning funnel upside down a few times. Solution was stored overnight at 4 °C. Aqueous phase (upper portion) was separated by boiling flask. Admixtures of acetone and chloroform were removed by rotary evaporation in vacuum at 38 °C.

The extracts encoded for MTT assay are as follows; RSNE: *Sambucus nigra* ethanol, RSNA: *Sambucus nigra* acetone; RVME: *Vaccinium myrtillus* ethanol; RVMA: *Vaccinium myrtillus* acetone; RACE: *Aronia melanocarpa* ethanol, RACA: *Aronia melanocarpa* acetone and RVCE: *Vaccinium corybosum* ethanol respectively.

### 2.4. LC-MS-IT-TOF

Identification and determination of the quantity of natural anthocyanins in selected plant matrices were performed by liquid chromatography on a reverse phase in conjunction with mass detection. Shimadzu LC-MS-IT-TOF 230 V CE (liquid chromatography on reverse phase coupling atmospheric pressure ionization with Ion-Trap (IT) and Time-of-Flight (TOF) technologies connected with mass detection) was used for qualitative and quantitative analysis of extracts for anthocyanins of these four fruits: *Vaccinium corymbosum*, *Vaccinium myrtillus*, *Sambucus nigra* and *Aronia melanocarpa*. Technology of hybrid mass spectrometer: ion trap (IT) in conjunction with mass analyzer time-of-flight (TOF) provides high-resolution mass spectra. It also allows to make fragmentation experiments (MSn, n = 1–10), which is the basis of structural analysis of unknown substances. The results of qualitative analysis by LC-MS-IT-TOF include high-resolution mass spectra (resolution

of 10,000), which is the molecular weight of each ion (the monoisotopic ion) measured to 4 decimal places of accuracy of weight measurement between 1–10 ppm. In general, mass spectra of anthocyanins obtained by electrospray ionization in positive mode contained the following signals: molecular ion (M)+, the ion of anthocyanidin ions representing a gradual breakdown of monosaccharide units—the neutral hexoses straight loss 162.0528 m/z respectively, pentosis with neutral straight loss 132.0423 m/z.

Samples for analysis were prepared by extraction with acetone and ethanol. The acetone extracts were purified before analysis on SPE cartridges (LiChrolut* RP-18 (40–63 um) 1000 mg/6 ml PP tubes, MERCK). The ethanol extracts were purified on sorbent Silica gel 100 C18, FLUKA. Ionization was performed using a conventional ESI source, in the positive ionization mode. The heat block and curved desolvation line (CDL) were maintained at 200 °C. Nitrogen was used as the nebulizing gas and drying gas, set at 1.5 L/min. The ESI source voltage was set at 4.5 kV and the detector voltage was set at 1.56 kV.

### 2.5. Lyofilisation Methods and Anthocyanin Stabilization

Freeze-drying process was done by the laboratory equipment GEA Lyophil SMART®SL 2 (GEA Lyophil GmbH, 92, Kaischeurener St., Hurth, Germany). Extracts were studied by the resistance measurements. The behavior of the products was studied while they were being cooled down to a temperature lower than −80 °C and subsequently reheated in an alcohol bath. The resistance and temperature of the extracts were measured during cooling and reheating. With this method it is possible to study the behavior of products while they are being frozen and heated. It is also possible to determine the temperature that the product must reach during freezing, as well as the product temperature that must be maintained during primary drying (sublimation temperature Tice). The measuring system used was AW 2 Eutectic monitor with a programmable TS 2 cooling and heating unit. The values measured were registered by the the DES software of the AW 2 eutectic monitor. The measurements can be valuated using the curves thus obtained.

### 2.6. Anti-Proliferative Activity

The human colorectal cancer cell line DLD1 was obtained from Prof. Dr. Ali Uğur URAL and cultured in 10% heat-inactivated fetal bovine serum (FBS), 1% penicillin-streptomycin supported medium RPMI 1640 medium. Cells were grown at 37 °C under 5% $CO_2$ conditions. DLD1 cells were then harvested and transferred to ELISA plates and no treatment was performed before 24 h. Anthocyanin samples were prepared with serum free medium and applied to cells at various concentrations (0.125–1 mg/mL) after then they were incubated for two-time intervals (24–48 h). Cell proliferation experiments were performed with the MTT assay. Viable tumor cells were counted for their ability to reduce yellow dye (MTT) to a blue formazan product [14]. Four hours later, the formazan product of MTT reduction was dissolved in isopropanol and the optical density of the plates was measured at 570 nm using an Elisa microplate reader. The data obtained from the MTT test were subjected to statistical analysis and at least three replicates were performed for each concentration, and the average data were taken into account. The statistical analysis was conducted using one-way analysis of variance (ANOVA) followed by Dunnett's test for comparison with control cells via SPSS version 23 for Mac OS (SPSS Inc., Chicago, IL, USA) and p less than 0.05 was selected as the level of significance.

### 2.7. Antimicrobial Activity

As test cultures, the following bacteria and yeasts from the American Type Culture Collection were used: *Candida albicans* ATCC 885-653; *Staphylococcus aureus* ATCC 25923; *Escherichia coli* ATCC 25922; *Enterococcus faecalis* ATCC 29212; *Streptococcus pyogenes* ATCC 19615. We also used clinical strains of bacteria and yeasts (*S. aureus*, *E. coli*, *S. pyogenes*, *C. albicans*) isolated from the oral cavities of patients suffering from inflammatory periodontium. The sensitivity of microorganisms to anthocyanins was determined by the agar diffusion test [15]. Anthocyanins were dissolved in DMSO (100 mg/mL). 100 µL bacterium inoculate

in the physiological solution were adjusted to the equivalent of 0.5 McFarland standard, and evenly spread on the surface of Muller-Hinton agar (incubated at $37 \pm 2\ ^{\circ}$C for 24 h); yeasts—on SDA agar (incubated at $35 \pm 2\ ^{\circ}$C for 48 h). 20 µL of extracts were introduced into wells 6 mm in diameter. The diameters of the inhibition zones were measured in millimetres including the diameter of the well. The antimicrobial effect was assessed by presence of growth inhibition zone. Each antimicrobial assay was performed at least three times.

As a positive control were used: gentamicin (10 mg/disk) for Gram-negative bacteria, ampicillin (10 mg/disk) for Gram-positive bacteria, nystatin (100 UI) for *Candida*. DMSO was used as negative control. For the analysis of the results of the experiments, we used statistical software Microsoft Office-Excel (2013) with the calculation of averages, error, and standard deviation.

To determine the minimum inhibitory concentration (MIC) and minimum bactericidal concentration (MBC), serial dilutions in Tryptone Soy Broth with the following anthocyanin concentrations were prepared: 100.0; 75.0; 50.0; 25.0; 20.0; 17.5; 15.0; 12.5; 10.0; 7,5; 5.0; and 2.5 mg/mL. The mixtures were then transposed into a 96-well microtitre (Greiner-BioOne, Austria), inoculated 100 µL an overnight inoculum of bacteria (0.5 McFarland standard) and incubated at $37 \pm 2\ ^{\circ}$C for 24 h; the yeasts – on SDA agar incubated at $35 \pm 2\ ^{\circ}$C for 48 h. The well contents were sown on the surface of Muller-Hinton agar.

## 3. Results

Freeze-drying is a process in which frozen raw materials are placed in a refrigerated vacuum system and directly dehydrated without thawing. The ice in the product is sublimed into water vapor. There is no deterioration in the cell structure during the freeze-drying process. The freeze-dried product also preserves the various properties of the raw material (shape, colour, nutritional value, flavor) better than other drying methods. It consists of four stages: sample pretreatment, deep freezing, primary drying and secondary drying. Genskowsky, et al. [16] defined freeze drying as a drying process through sublimation. Freeze-drying is considered one of the best methods of preserving the organoleptic and nutritional properties of biological products [17]. Freeze-dried products are characterized by low water activity, low changes in volume and shape, high rehydration capacity, increased porosity and presenting a glassy state [18].

### 3.1. Qualitative Anthocyanin Content by LC-MS-IT-TOF Analyses

The results of the LC-MS-IT-TOF analyses clearly show that the ethanolic extracts contain more anthocyanins than acetone extracts. 25 anthocyanins were detected in the ethanol extract and 19 in the acetone extract. The LC-MS-IT-TOF profiles showed the advantage of using ethanol as the solvent for extraction from berry fruits.

In our study, cyanidine-3-glukoside is a common compound in the content of all studied samples. We also detected cyanidine-3-galactoside, malvidine-3-galactoside, malvidine-3-glucoside, cyanidine-3,5-diglucoside and cyanidine-3-rutinoside in various amounts, respectively. Cyanidine-3,5-diglucoside, cyanidin-3-rutinoside and, most importantly, cyanidine-3-sambusioside were only found in *Sambucus nigra* samples. In addition, malvidine-3-galactoside and malvidin-3-glucoside were detected only in *Vaccinium* species. The anthocyanin contents of lyophilizates after their ethanolic and acetone extraction and identification by LC-MS-IT-TOF are given in Tables 1–4.

**Table 1.** Anthocyanin contents of elderberry lyophilizates in regard to ethanolic fruit extract (●: presence; -: absence).

| Compound Assigned | Rt (min) | Molecular Ions | Fragment Ions | Bilberry *Vaccinium myrtillus* | Highbush Blueberry *Vaccinium corymbosum* | Eldelberry *Sambucus nigra* | Black Chokeberry *Aronia melanocarpa* |
|---|---|---|---|---|---|---|---|
| cyanidine-3,5-diglucoside | 8,985 | 611,616 | 449,112; 287,06 | - | - | ● | - |
| cyanidine-3-sambubioside-5-glukoside | 9,390 | 743,202 | 581,158; 449,106; 287,056 | - | - | ● | - |
| delfinidine + hexose | 9,545 | 465,089 | 303,042 | - | ● | - | - |
| delfinidine-3-galactoside | 9,565 | 465,102 | 303,051 | ● | - | - | - |
| delfinidin-3-glucoside | 10,005 | 465,102 | 303,051 | ● | - | - | - |
| cyanidine-3-galactoside | 10,300 | 449,106 | 287,055 | ● | ● | - | ● |
| delfinidine-3-arabinoside | 10,375 | 435,093 | 303,050 | ● | ● | - | - |
| cyanidine-3-glucoside | 10,677 | 449,106 | 287,055 | ● | ● | ● | ● |
| cyanidine-3-sambubiozid | 10,750 | 581,148 | 287,053 | - | - | ● | - |
| petunidine-3-galaktoside | 10,820 | 479,117 | 317,066 | ● | - | - | - |
| petunidine + hexoze | 10,800 | 479,104 | 317,056 | - | ● | - | - |
| cyanidine-3-arabinoside | 10,970 | 419,098 | 287,055 | ● | ● | - | ● |
| cyanidine-3-rutinoside | 11,072 | 595,164 | 449,108; 287,050 | - | - | ● | - |
| petunidine-3-glucoside | 11,105 | 479,118 | 317,066 | ● | - | - | - |
| pelargonidíne-3-glucoside | 11,260 | 433,113 | 271,060 | - | - | ● | - |
| peonidine-3-galactoside | 11,360 | 463,120 | 301,070 | ● | - | - | - |
| peonidine + hexoze | 11,365 | 463,110 | 301,060 | - | ● | - | - |
| petunidine3-arabinoside | 11,420 | 449,108 | 317,067 | ● | ● | - | - |
| pelargonidíne-3-sambubioside | 11,485 | 565,133 | 271,046 | - | - | ● | - |
| peonidine-3-glucoside | 11,690 | 463,124 | 301,070 | ● | - | - | - |

**Table 1.** *Cont.*

| Compound Assigned | Rt (min) | Molecular Ions | Fragment Ions | Bilberry *Vaccinium myrtillus* | Highbush Blueberry *Vaccinium corymbosum* | Eldelberry *Sambucus nigra* | Black Chokeberry *Aronia melanocarpa* |
|---|---|---|---|---|---|---|---|
| malvidininne-3-galactoside | 11,704 | 493,133 | 331,082 | ● | ● | - | - |
| cyanidine-3-xyloside | 11,815 | 419,097 | 287,053 | - | - | - | ● |
| peonidine-3-arabinoside | 11,965 | 433,100 | 301,060 | ● | ● | - | - |
| malvidine-3-glucoside | 11,955 | 493,133 | 331,082 | ● | ● | - | - |
| malvidine-3-arabinoside | 12,265 | 463,124 | 331,080 | ● | ● | - | - |

**Table 2.** Anthocyanin contents of elderberry lyophilizates in regard to acetone fruit extract (●: presence; -: absence).

| Compound Assigned | Rt (min) | Molecular Ions | Fragment Ions | Bilberry *Vaccinium myrtillus* | Highbush Blueberry *Vaccinium corymbosum* | Eldelberry *Sambucus nigra* | Black Chokeberry *Aronia melanocarpa* |
|---|---|---|---|---|---|---|---|
| cyanidine-3,5-diglucoside | 9,305 | 611,160 | 449,112; 287,060 | - | - | ● | - |
| cyanidine-3-sambubioside-5-glucoside | 9,835 | 743,202 | 581,158; 449,106; 287,056 | - | - | ● | - |
| delfinidine + hexoze | 10,015 | 465,101 | 303,050 | ● | ● | - | - |
| cyanidine-3-sambubioside | 11,050 | 581,148 | 287,054 | - | - | ● | - |
| cyanidine-3-galactoside | 10,671 | 449,106 | 287,050 | ● | ● | - | ● |
| delfinidine-3-arabinoside | 10,705 | 435,091 | 303,050 | ● | ● | - | - |
| cyanidín-3-glukoside | 10,995 | 449,106 | 287,050 | ● | ● | ● | ● |
| petunidine + hexose | 11,129 | 479,116 | 317,065 | ● | ● | - | - |
| cyanidine-3-arabinoside | 11,265 | 419,098 | 287,054 | ● | ● | - | ● |
| cyanidine-3-rutinoside | 11,315 | 595,164 | 449,108; 287,050 | - | - | ● | - |
| pelargonidíne-3-glucoside | 11,559 | 433,113 | 271,060 | - | - | ● | - |
| peonidine + hexose | 11,660 | 463,120 | 301,070 | ● | ● | - | - |

**Table 2.** *Cont.*

| Compound Assigned | Rt (min) | Molecular Ions | Fragment Ions | Bilberry *Vaccinium myrtillus* | Highbush Blueberry *Vaccinium corymbosum* | Eldelberry *Sambucus nigra* | Black Chokeberry *Aronia melanocarpa* |
|---|---|---|---|---|---|---|---|
| petunidine-3-arabinoside | 11,685 | 449,109 | 317,066 | ● | ● | - | - |
| pelargonidíne-3-sambubio-side | 11,725 | 565,155 | 271,060 | - | - | ● | - |
| malvidine-3-galactoside | 11,973 | 493,131 | 331,079 | ● | ● | - | - |
| cyanidine-3-xyloside | 12,035 | 419,097 | 287,053 | - | - | - | ● |
| peonidine-3-arabinoside | 12,215 | 433,111 | 301,070 | ● | ● | - | - |
| malvidine-3-glucoside | 12,240 | 493,131 | 331,079 | ● | ● | - | - |
| malvidine-3-arabinoside | 12,524 | 463,122 | 331,079 | ● | ● | - | - |

**Table 3.** Quantitative anthocyanin content of ethanol extracts by LC-MS-IT-TOF analyses [ppm]. (-: absence).

| Compound Assigned | Bilberry *Vaccinium myrtillus* | Highbush Blueberry *Vaccinium corymbosum* | Eldelberry *Sambucus nigra* | Black Chokeberry *Aronia melanocarpa* |
|---|---|---|---|---|
| cyanidine-3-glucoside | 153.04 ± 29.64 | 1.39 ± 0.29 | 240.51 ± 50.87 | 52.52 ± 2.60 |
| cyanidine-3-galactoside | 148.91 ± 30.50 | 35.42 ± 1.03 | - | 356.83 ± 84.76 |
| malvidine-3-galactozide | 43.19 ± 8.87 | 65.46 ± 11.74 | - | - |
| malvidine-3-glucoside | 144.75 ± 31.03 | 3.64 ± 0.29 | - | - |
| cyanidine-3,5-diglucoside | - | - | 46.44 ± 2.78 | - |
| cyanidine-3-rutinozide | - | - | 8.96 ± 0.58 | - |

### 3.2. MTT Assay Results

DLD1 human colorectal cancer cell line was used to determine the antiproliferative effects of different berry extracts. MTT analysis shows that the viability of DLD1 cells is significantly reduced by applying extracts at different time intervals. The results show that all anthocyanins have time and dose-dependent inhibitory effect on the viability of cancer cells ($p < 0.05$). Responses of DLD1 cells to increased anthocyanin concentrations were given in Figures 1 and 2.

Indications: RSNE: *Sambucus nigra* ethanol, RSNA: *Sambucus nigra* acetone; RVME: *Vaccinium myrtillus* ethanol; RVMA: *Vaccinium myrtillus* acetone; RACE: *Aronia melanocarpa* ethanol, RACA: *Aronia melanocarpa* acetone and RVCE: *Vaccinium corybosum* ethanol

**Table 4.** Quantitative anthocyanin content of acetone extracts by LC-MS-IT-TOF analyses [ppm]. (-: absence).

| Compound Assigned | Bilberry *Vaccinium myrtillus* | Highbush Blueberry *Vaccinium corymbosum* | Eldelberry *Sambucus nigra* | Black Chokeberry *Aronia melanocarpa* |
|---|---|---|---|---|
| cyanidine-3-glucoside | 550.60 ± 18.37 | 1.48 ± 0.09 | 958.47 ± 139.68 | 73.97 ± 5.65 |
| cyanidine-3-galactoside | 545.89 ± 37.50 | 45.57 ± 1.94 | - | 1103.49 ± 10.99 |
| malvidine-3-galactoside | 140.95 ± 5.24 | 250.49 ± 21.70 | - | - |
| malvidine-3-glucoside | 456.42 ± 23.98 | 13.88 ± 2.54 | - | - |
| cyanidine-3,5-diglucoside | - | - | 198.08 ± 3.88 | - |
| cyanidine-3-rutinoside | - | - | 21.62 ± 2.65 | - |

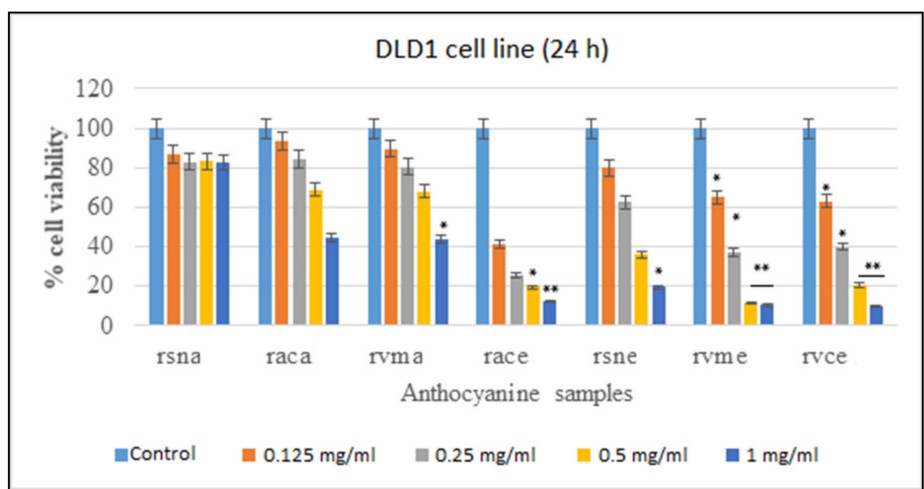

**Figure 1.** The antiproliferative effects of anthocyanines for 24 h application (* $p < 0.05$; ** $p < 0.001$).

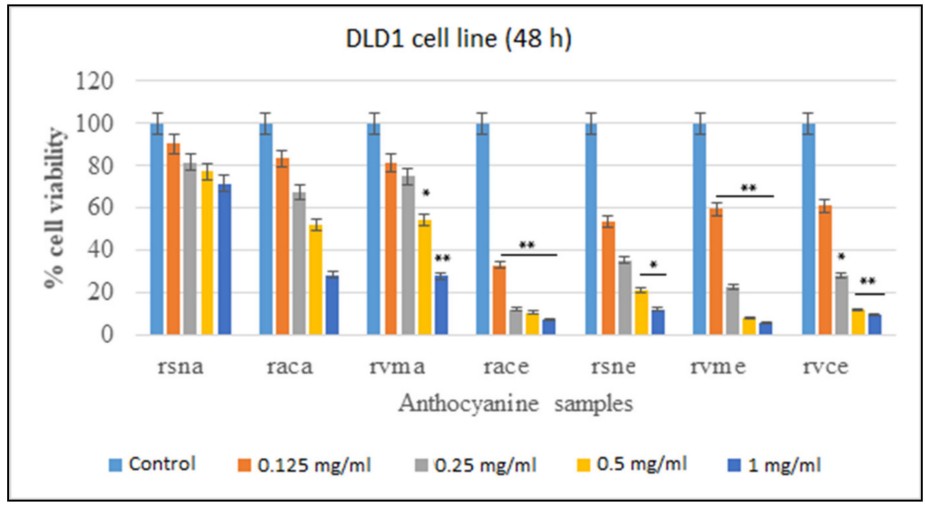

**Figure 2.** The antiproliferative effects of anthocyanines for 48 h application (* $p < 0.05$; ** $p < 0.001$).

When we consider MTT results in general, it can be said that the ethanolic extract of *Aronia melanocarpa* is more effective than others. This is followed by *Vaccinium myrtillus* and *Vaccinium corybosum* etanolic extracts, respectively. As a general contribution, we can clearly say that ethanol extracts have a much better effect than acetone extracts for an antiproliferative effect. The $IC_{50}$ doses of the extracts were given Table 5.

**Table 5.** The $IC_{50}$ doses of anthocyanins on DLD1 Cells.

| $IC_{50}$ | RSNA | RACA | RVMA | RACE | RSNE | RVME | RVCE |
|---|---|---|---|---|---|---|---|
| 24 h | up to 1mg/mL | 0.88 mg/mL | 0.875 mg/mL | 0.108 mg/mL | 0.370 mg/mL | 0.175 mg/mL | 0.196 mg/mL |
| 48 h | up to 1mg/mL | 0.541 mg/mL | 0.576 mg/mL | 0.093 mg/mL | 0.151 mg/mL | 0.159 mg/mL | 0.167 mg/mL |

Indications: RSNE: *Sambucus nigra* ethanol, RSNA: *Sambucus nigra* acetone; RVME: *Vaccinium myrtillus* ethanol; RVMA: *Vaccinium myrtillus* acetone; RACE: *Aronia melanocarpa* ethanol, RACA: *Aronia melanocarpa* acetone and RVCE: *Vaccinium corybosum* ethanol.

### 3.3. Antimicrobial Activity of Anthocyanins

The results of the research show that anthocyanins of *Aronia melanocarpa* and *Vaccinium myrtillus* display antimicrobial activity against clinical and reference strains of *S.aureus*. Only *Vaccinium myrtillus (acetone)* was shown to exert antimicrobial activity against reference and clinical strains of *E. faecalis*.

Antimycotic activity of anthocyanins was not revealed. Absence of antimicrobial activity of anthocyanins against *S. pyogenes* and clinical strains of *E. coli* was observed.

It was further demonstrated that anthocyanins of *Sambucus nigra* did not show any antimicrobial activity. It was proved that *Vaccinium myrtillus (acetone)* displays antimicrobial activity against clinical and reference strains of *S. aureus* and *E. faecalis*.

The antimicrobial activities of extracts against typical and clinical strains were given at Tables 6 and 7 respectively.

**Table 6.** Antimicrobial activity of anthocyanins against typical strains, zones inhibition in millimeters including diameter of well, mm (n = 3, x ± SD). ( − : no inhibition).

| № | Samples | *S. aureus* ATCC 25923 | *E. coli* ATCC 25922 | *E. faecalis* ATCC 29212 | *S. pyogenes* ATCC 19615 | *C. albicans* ATCC 885-653 |
|---|---|---|---|---|---|---|
| 1 | *Vaccinium myrtillus* (ethanol) | 9.33 ± 0.50 | - | - | - | - |
| 2 | *Vaccinium myrtillus* (acetone) | 9.50 ± 0.50 | 8.66 ± 0.58 | 17.66 ± 0.58 | - | |
| 3 | *Aronia melanocarpa* (ethanol) | 16.50 ± 0.29 | 8.33 ± 0.58 | - | - | |
| 4 | *Aronia melanocarpa* (acetone) | 13.33 ± 0.58 | - | 12.83 ± 0.58 | - | - |
| 5 | *Sambucus nigra* (ethanol) | - | - | - | - | - |
| 6 | *Sambucus nigra* (acetone) | - | - | - | - | - |
| 7 | *Vaccinium corymbosum* (ethanol) | - | 10.83 ± 0.76 | - | - | - |

**Table 7.** Antimicrobial activity of anthocyanins against clinical strains, zones inhibition in millimeters including diameter of well, mm (n = 3, x ± SD). (− : no inhibition).

| № | Samples | *S. aureus* | *E. coli* | *E. faecalis* | *S. pyogenes* | *C. albicans* |
|---|---------|-------------|-----------|---------------|---------------|---------------|
| 1 | *Vaccinium myrtillus* (ethanol) | 9.11 ± 0.29 | - | - | - | - |
| 2 | *Vaccinium myrtillus* (acetone) | 12.33 ± 0.58 | - | 11.33 ± 0.58 | - | - |
| 3 | *Aronia melanocarpa* (ethanol) | 13.00 ± 1.00 | - | - | - | - |
| 4 | *Aronia melanocarpa* (acetone) | 10.66 ± 0.58 | - | - | - | - |
| 5 | *Sambucus nigra* (ethanol) | - | - | - | - | - |
| 6 | *Sambucus nigra* (aceton) | - | - | - | - | - |
| 7 | *Vaccinium corymbosum* (ethanol) | 10.83 ± 0.76 | - | - | - | - |

The MIC and the MBC of *Vaccinium myrtillus* (ethanol) against clinical and reference isolates of *S. aureus* equaled to 15.0 and 20.0 mg/mL, respectively. The antimicrobial activity by MIC (2.5 mg/mL) and MBC (5.0 mg/mL) indices of *Vaccinium myrtillus* (acetone) that was determined against the reference *E. faecalis* ATCC 29212, equaled to 10.0 and 15.0 mg/mL, respectively, against the clinical isolates of *Enterococci.* The antimicrobial activity of *Vaccinium myrtillus* (acetone) was observed upon *staphylococcus* isolates: MIC was 10.0-15.0 mg/mL, and MBC was 20.0 mg/mL. No bactericidal effect of this extract upon the other isolates was observed. MIC against typical *E. coli* equaled to 25.0 mg/mL. *Aronia melanocarpa* (ethanol) exerted a bactericidal effect upon clinical isolates of *S. aureus* (MBC was 10.0 mg/mL, and MIC was 5.0 mg/mL). *Aronia melanocarpa* (acetone) had a lower activity against clinical isolates of *S. aureus* (MBC equaled to 15.0 mg/mL; MIC equaled to 12.5 mg/mL). At the same time, its antimicrobial effect upon the reference *E. faecalis* was shown (MBC equaled to 10.0 mg/mL, and MIC equaled to 15.0 mg/mL). *Vaccinium corymbosum* (ethanol) also had bactericidal effect upon *S. aureus* and the reference isolates of *E. coli* (MBC equaled to 15.0 mg/mL; MIC equaled to 12.5 mg/mL).

## 4. Discussion

Anthocyanins are natural phytonutrients that have beneficial effects on animals and humans. Revealing the anthocyanin content of herbal preparations is very important both in nutritional and pharmacological terms. According to our results, we can clearly say that *Vaccinium* species possessed the highest number of anthocyanins. Anthocyanin content may vary between different plant sources; especially berry fruits are very rich in different polyphenols, including anthocyanins. The anthocyanin content of individual fruits may also be affected by environmental factors such as fruit maturity at harvest [19]. Moreover, the quantity of anthocyanins can be affected by process conditions. Our study confirms, as has been shown in research that the freeze-drying method is the best method to protect phenolics and anthocyanins, it has been reported in many studies before [20,21]. Freeze drying is a useful method to preserve color, flavor, and nutrient compounds due to the lack of water, low pressure and temperature [22].

Anthocyanins are a water-soluble flavonoid class that exhibits a number of pharmacological effects, such as the prevention of various diseases such as cancer. Potential anti-cancer effects are reported to be based on a wide range of biological activities such as antioxidant and anti-mutagenesis effects, inhibition of cell proliferation, induction of cell cycle arrest and apoptosis.

Anthocyanins have been studied in detail for their anticancer properties based on in vitro studies and animal models. Therefore preventing tumor growth, antiangiogenesis is a process that prevents the formation of new blood vessels that supply oxygen to tumor cells. Like many other phytochemicals, flavonoids and anthocyanins are potential anticancer and antiangiogenic agents [23]. Anticancer effects of anthocyanins from different herbal sources have been studied in many types of cancer, such as esophagus, colon, breast, liver, hematological and prostate. Findings from a previous study indicate that F344 rats bound to N-nitrozo methyl benzyl amine have a chemopreventive potential in freeze-dried black raspberry and anthocyanin-rich fraction [24]. Colorectal cancer is the second most common cause of cancer death and affects more than one million patients worldwide each year [25]. Anthocyanins have appeared as hopeful compounds that may promote their health benefits in colorectal cancer due to their antioxidant and anti-inflammatory properties [26,27]. In addition, the protective effects of anthocyanins on colorectal cancer have been reported in previous studies [28,29].In this study, we evaluated whether there is an effect on proliferation after cancer occurs. For this purpose, we chose the DLD1 human colorectal cell line. Moreover, we aimed to reveal the antimicrobial effects of anthocyanins as well as their antiproliferative effects. In the present study, anthocyanins from the freeze-drying process were shown to inhibit the proliferation of colon cancer cells, consistent with previous reports [30–32]. It has been reported that the anthocyanins-rich extracts from Chinese blueberry suppressed the proliferation of colon carcinoma cell lines, DLD1 and COLO-205 cells [33]. Another study reported that cyanidine and anthocyanins-rich extracts obtained from tart cherry were able to induce a dose-dependent decrease in cell proliferation both in HCT-116 and HT-29 cells [34].

Polyphenolic compounds, including anthocyanins, have antimicrobial activity against a wide variety of microorganisms, especially in the growth inhibition of pathogens [35]. Anthocyanins execute their antimicrobial activities by stimulating cell damage through various mechanisms and then triggering cell destruction. Antimicrobial activity of plant phenolic compounds against human pathogens has been broadly studied to qualify and develop new useful food contents as well as pharmaceutical products. Although there is limited information on this subject, some studies have been performed. Burdulis, et al. [36] determined the total anthocyanin content in blueberry (*Vaccinium myrtillus*) and blueberry (*Vaccinium corymbosum*) and identify the antimicrobial properties of their extracts. They reported that the extracts showed inhibitory effects on the growth of both Gram-positive and Gram-negative strains and while *C. freundii* and *E. faecalis* strains were the most susceptible, *E. coli* showed the greatest resistance among the tested bacteria. In another study, it was reported that European cranberry (*Vaccinium macrocarpon*) extracts inhibited the growth of a wide range of human pathogenic bacteria with *L. monocytogenes* and *E. faecalis*, *S. enterica* ser. *Typhimurium*, and *S. aureus* were found to be of moderate resistance and *E. coli* rods were the least sensitive [37]. Genskowsky, Puente, Pérez-Álvarez, Fernández-López, Muñoz and Viuda-Martos [17] reported maqui berry extracts had an antibacterial activity with the highest sensitivity to *Listeria innocua* and *Aeromonas hydrophilia*. These antimicrobial activities of anthocyanin-containing extracts are possible due to the multiple mechanisms and synergistic effects of various phytochemicals in the extracted content [38]. Therefore, it is necessary to investigate the content in detail. The results we obtained in our study are compatible with previous studies.

## 5. Conclusions

The unique and optimal technique for anthocyanin extractions and their preservation by lyophilization were presented in this study. Anthocyanins are instable in higher temperature and lower pH and our process of freeze-drying stabilizes them. The final product of anthocyanins in our R&D could be used as food supplement, gelatin tablets, but also injections. They have been suggested to be beneficial for cardiovascular and neurodegenerative disease, as well as eye, muscle disorders [39] and sarcopenia, even for possible dietary treatment of Duchenne muscular dystrophy [40]. By removing the water

from the berry fruit extracts and sealing the clear anthocyanins in vials, the product can be stored easily and longer, as well as shipped regardless of the ambient conditions. Until now, pharmaceutical companies often use freeze-drying to increase the shelf life of products, such as vaccines and other injectable.

However, much more studies are needed to determine the true effects of anthocyanins on these health-promoting properties [41]. Studies to date have shown numerous benefits arising from including berry fruits and phytonutrients (such as anthocyanins) in the daily diet. Like other herbs and naturally sourced medicinal products, berry fruits and extracts require extensive research in humans to determine its efficacy, safety, and mechanisms of action. Our data suggest that the anthocyanin content in the extracts obtained by the freeze-drying method was quite good, and some extracts were found to have a strong antiproliferative effect on colorectal cancer cells and have an antibacterial effect on typical and clinical strains.

**Author Contributions:** I.S. devised the project, the main conceptual ideas and proof outline. He is one of the authors of patent number 288313/2015: "*Method of Anthocyanins Isolation from Berry Fruits of Medicinal Plants and their Lyophilisation*" certificated by the Industrial Property Office of the Slovak Republic in Banska Bystrica. P.L. worked out almost all of the technical details, and performed the numerical calculations for the suggested experiment. M.K. and E.N.Ş.S. processed the experimental data, performed the biological analysis, drafted the manuscript and designed the figures. All authors discussed the results and commented on the manuscript. All authors have read and agreed to the published version of the manuscript.

**Funding:** This study was supported by the Ministry of Education, Science, Research and Sport of the Slovak Republic, the project: 00162-0001 (MS SR-3634/2010-11) "*Isolation of Natural Plant Substances by Lyophilisation and Change of their Qualitative-Quantitative Properties*".

**Institutional Review Board Statement:** Not applicable for studies not involving humans or animals.

**Informed Consent Statement:** Not applicable for studies not involving humans.

**Data Availability Statement:** Data openly available in a public repository that issues datasets with DOIs.

**Conflicts of Interest:** The authors declare that the research was conducted in the absence of any commercial or financial relationships that could be construed as a potential conflict of interest.

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
