# Peer review of "Antiproliferative and Antimicrobial Activity of Anthocyanins from Berry Fruits after Their Isolation and Freeze-Drying"

_applsci, doi:10.3390/app11052096_

Round 1
Reviewer 1 Report
Regarding the manuscript entitled ‘Antiproliferative and antimicrobial activity of anthocyanins after their isolation and freeze-drying’. The concept of the manuscript is novel, fits and suitable to publish in Applied Sciences Journal. This manuscript is generally well written and clearly presented however still need to address many comments and thus require substantial major revision before its acceptance.
- Provide a nice graphical abstract representing the overview of the MS with key highlights.
- Title need to modify which can describe whole research work. Mention fruits name or berry fruits in the title so modify accordingly Keywords should be according to results
- Authors have used fruits for anthocyanin extraction so mention fruits instead of plants materials throughout the manuscript.
- In abstract authors should mention the values of results and importance of research work in one or two sentences. In abstract give full form of the abbreviations used.
- Also mention somewhere the full form of abbreviations used in the manuscript.
- In the introduction section, write the novelty of the work and the problem statement clearly. The defined research objectives should be mentioned ta the end of introduction. Substantial discussion about antibacterial and anticancer studies are essential.
- To understand the antibacterial and anticancer studies authors should study the antioxidant activities for this pl refer and cite Nanomaterials 10 (8), 1457, 2020; Environmental Science and Pollution Research 25 (11), 10250-10263, 2018. Why you choose DLD1 cell lines for anticancer studies need to mention somewhere.
- Importance and advantages of this studies in medical field need to describe somewhere.
- Write the practical applications and future research perspectives and challenges by adding a new section before conclusions
- The conclusion of the study is not discussed with the specific output obtained from the study, it could be modified with precise outcomes with a take home message.
- English and grammar mistakes are present. The author should check the manuscript by native English Speaker to improve the quality of the manuscript.
Author Response
Dear Sir,
Thank you very much for your supporting the publication of our manuscript dedicated to the unique technology of isolation and especially stabilization of anthocyanins from plant fruits and their selected biological effects. As the main author of the manuscript, I was waiting for a suitable journal and topic - Recent Advances in the Ecological Extraction and Application of Bioactive Plant Compounds.
At the same time, my thanks are for your expert review with your comments on the content of the text and its stylization. We tried to respond to all your comments and remarks with our co-authors.
I was personally concerned about all the comments regarding the detailed description of lyophilization within the material and methods. I have added these facts to the text.
Due to the fact, it is a unique method of isolation and especially stabilization of anthocyanins. They are very sensitive to all environmental conditions, which influence to their decomposition. It is very important to point out the possibilities of using pure, characterized anthocyanins for human health. We currently work with several laboratories around the world, in addition to co-authors, also with the American University of Illinois in Chicago. I have added current information in conclusion about the experiments to the text.
I am sending you a revised manuscript recorded in an attachment.
Dear Sir,
Thank you very much for your supporting the publication of our manuscript dedicated to the unique technology of isolation and especially stabilization of anthocyanins from plant fruits and their selected biological effects. As the main author of the manuscript, I was waiting for a suitable journal and topic - Recent Advances in the Ecological Extraction and Application of Bioactive Plant Compounds.
At the same time, my thanks are for your expert review with your comments on the content of the text and its stylization. We tried to respond to all your comments and remarks with our co-authors.
I was personally concerned about all the comments regarding the detailed description of lyophilization within the material and methods. I have added these facts to the text.
Due to the fact, it is a unique method of isolation and especially stabilization of anthocyanins. They are very sensitive to all environmental conditions, which influence to their decomposition. It is very important to point out the possibilities of using pure, characterized anthocyanins for human health. We currently work with several laboratories around the world, in addition to co-authors, also with the American University of Illinois in Chicago. I have added current information in conclusion about the experiments to the text.
I am sending you a revised manuscript recorded in an attachment.

Reviewer 2 Report
The manuscript titled "Antiproliferative and antimicrobial activity of anthocyanins after their isolation and freeze-drying" concerns the method of fruit processing in the context of anthocyanins content as well as their antiproliferative and antimicrobial effects.
The anthocyanins are valuable bioactive ingredients with a proved positive influence on health.
The authors conducted and described an interesting experiment but the text needs a few correction and clarification.
Because the lines in the file are not numbered, I will refer to pages.
In my opinion, the first paragraph of the introduction should address the applied science aspect and the general role of the processing food which contains plenty of phytochemicals.
In the purpose of the study preparation of the powder as a food supplement is mentioned. However, in the methodology, results and discussion, the issue is not taken. What is more, two methods of extraction are examined. Please clarify the goal.
Regarding the freeze-drying, the reader has the impression that the authors investigated the effect of freeze-drying, but the methodology does not include an appropriate description. According to information on page 2, fresh plant material was conventionally frozen. Please clarify the method of freezing researched.
Page 2: Chemicals and all mentioned laboratory reagents - The detailed information about producer should be added (city, country).
Page 5 and all the next with tables: the footers are needed and explanation what dot of dash mean.
Figure 1 - 2 and Table 5: please explain what "rsna3I, et cetera" mean.
Table 6: In my opinion, the inhibition zone should be calculated without the diameter of well.
The results and discussion should be modified, when the aim will be clarified.
The conclusions do not relate to the purpose of the work. We do not know if it has been achieved. Please clarify.
Author Response
Dear Sir,
Thank you very much for your supporting the publication of our manuscript dedicated to the unique technology of isolation and especially stabilization of anthocyanins from plant fruits and their selected biological effects. As the main author of the manuscript, I was waiting for a suitable journal and topic - Recent Advances in the Ecological Extraction and Application of Bioactive Plant Compounds.
At the same time, my thanks are for your expert review with your comments on the content of the text and its stylization. We tried to respond to all your comments and remarks with our co-authors.
I was personally concerned about all the comments regarding the detailed description of lyophilization within the material and methods. I have added these facts to the text.
We have provided sufficient information about chemicals and their manufacturers.
In connection with microbiological methods for testing anthocyanins, we used the standard method. It was used to test the microbiological activity of many essential oils of medicinal plants, about which scientific articles have been published in professional journals in Slovakia, but also abroad.
The discussion is conducted in the levels of comparing information directly with the process of extraction and completion of the lyophilization method. Lyophilization of natural substances - anthocyanins was developed and used as a stabilization method for the first time in the world (the presented method is patented at the Industrial Property Office of the Slovak Republic).
It is very important to point out the possibilities of using pure, characterized anthocyanins for human health. We currently work with several laboratories around the world, in addition to co-authors, also with the American University of State Illinois in Chicago. I have added current information of our R&D in conclusion about the experiments to the text.
I am sending you a revised manuscript recorded in an attachment.

Round 2
Reviewer 1 Report
In the revised manuscript authors have not addressed all raised comments which is surprising. Need precise response for following comments in the revised version of the manuscript.
- In the introduction section, substantial discussion about antibacterial and anticancer studies are essential.
- To understand the antibacterial and anticancer mechanism authors should study the antioxidant activities.
- In antibacterial studies only measurement of ZOI is not sufficient authors should study MIC and MBC values for each selected strain.
Author Response
Acknowledgments, support and interest in the presented issues by the reviewer
Thank you very much for re-checking and commenting our manuscript of the scientific article. In relation to individual comments and remarks, the following can be stated:
- information and knowledge on antibacterial and anticancer properties from studies have been added,
-antioxidation activities in our case were monitored by colleagues from our university and in cooperation with abroad. Information and links to individual scientific studies are provided. Unfortunately, the results of the studies were contradictory, especially with regard to a number of factors affecting the plant material used, the isolation of the extracts themselves and the various methods of antioxidant activity used,
-Yes, ZOI was used in antibacterial tests, which is a standard method used in a number of published works (excluding MIC and MBC), an example is the article from 2020: Chemical Composition, Algicidal, Antimicrobial, and Antioxidant Activities of the Essential Oil of Taiwania flousiana Gauseen, by Hongmei Liu et all, Molecules, 2020, 25, 967

Reviewer 2 Report
The revised manuscript "Antiproliferative and antimicrobial activity of anthocyanins after their isolation and freeze-drying" is significantly improved by the authors.
I just have a suggestion to repeat the explanation of the extracts abbreviations in Figure 2 and Table 5.
Author Response
Acknowledgments, support and interest in the presented issues by the reviewer:
Thank you very much for re-checking and commenting on our manuscript of the scientific article. In relation to individual comments and remarks, the following can be stated:
-adjustments to the figure and table according to the comments have been made.
I appreciate our very useful cooperation in revising the manuscript, which brings interesting information, results and knowledge for readers with their significant use in practice and new innovative technologies.

Round 3
Reviewer 1 Report
In the revised manuscript authors have not addressed all raised comments which is surprising. Need precise response for following comments in the revised version of the manuscript.
- Very contradictory response from the authors. To understand the mechanisms of anticancer and antibacterial activity determination of antioxidant activity is quiet essential. Yes of course everyone knows the biogenic activities of any phytochemical depends on the type plants genotype, soil-climatic properties, antioxidant assay and isolation procedure. Provide your obtained Antioxidant studies results.
- I never said that ZOI is not used to determine antibacterial effect however to understand the exact mechanism and quantitative information, MIC and MBC is essential so authors should study MIC and MBC values for each selected strain.
Author Response
Dear Reviewer,
I appreciate your efforts, comments and notes to the content and text of our manuscript in the original scientific work. Based on your comments, we made and supplemented the results of antioxidant activity and other microbiological tests (MIC, MBC) were performed in the laboratory.
Please, see my attachment.
Thank you for your support and the opportunity to publish in the journal Applied Sciences.

Round 4
Reviewer 1 Report
The concentration of anthocyanin used for MIC and MBC studies is not correct. Pl check also mention what concentration authors have used for ZOI in Table 6 and Table 7 for each microbial strain?
Author Response
- Our results easy to compare with other authors.
In papper: Yong Ma, Sujuan Ding, Yanquan Fei, Gang Liu, Hongmei Jang, Jun Fang, Antimicrobial activity of anthocyanins and catechins against foodborne pathogens Escherichia coli and Salmonella, Food Control, Volume 106, 2019, 106712, ISSN 0956-7135
The minimum inhibitory concentration (MIC) of anthocyanins against E.coli and Salmonella is 10–400 mg/ml. The MIC of catechins against E.coli and Salmonella is 6–50 mg/ml.
In other article:
Antimicrobial effect of extract of fruits and leaves of Vaccinium myrtillus L. were studied. In this study the extracts performed on 30 clinical isolates, including strains of Escherichia coli, Enterococcus faecalis and Proteus vulgaris. The values for MIC were in the range from 5 to 40 mg/ml.
Dragana M. Vučić, Miroslav R. Petković, Branka B. Rodić-Grabovac, Olgica D. Stefanović, Sava M. Vasić and Ljiljana R. Čomić
Antibacterial and antioxidant activities of bilberry (Vaccinium myrtillus L.) in vitro
Available from: African Journal of Microbiology Research Vol. 7(45), pp. 5130-5136, 14 November, 2013 DOI: 10.5897/AJMR2013.2524 ISSN 1996-0808
- For agar diffusion method anthocyanins were dissolved in DMSO (100 mg/ml). We put in 20 µL of anthocins were introduced into wells 6 mm in diameter.
